# The Context-Dependent Impact of Integrin-Associated CD151 and Other Tetraspanins on Cancer Development and Progression: A Class of Versatile Mediators of Cellular Function and Signaling, Tumorigenesis and Metastasis

**DOI:** 10.3390/cancers13092005

**Published:** 2021-04-21

**Authors:** Sonia Erfani, Hui Hua, Yueyin Pan, Binhua P. Zhou, Xiuwei H. Yang

**Affiliations:** 1Department of Pharmacology and Nutritional Sciences, College of Medicine, University of Kentucky, Lexington, KY 40536, USA; sonia.erfani@stelizabeth.com; 2Markey Cancer Center, University of Kentucky Medical Center, Lexington, KY 40536, USA; 3Pharmacy Department, St. Elizabeth Healthcare, Edgewood, KY 41017, USA; 4The First Affiliated Hospital of University of Science and Technology of China, Hefei, Anhui 230001, China; huahui@mail.ustc.edu.cn (H.H.); panyueyin@ustc.edu.cn (Y.P.); 5Provincial Hospital, Hefei, Anhui 230001, China; 6Department of Molecular and Cellular Biochemistry, College of Medicine, University of Kentucky, Lexington, KY 40536, USA; peter.zhou@uky.edu

**Keywords:** CD151, tetraspanins, breast cancer, Wnt pathway, integrins, cancer stem cells, exosome, epithelial-mesenchymal transition (EMT)

## Abstract

**Simple Summary:**

Tetraspanins are a family of molecules abundantly expressed on the surface of normal or tumor cells. They have been implicated in recruiting or sequestering key molecular regulators of malignancy of a variety of human cancers, including breast and lung cancers, glioblastoma and leukemia. Yet, how their actions take place remains mysterious due to a lack of traditional platform for molecular interactions. The current review digs into this mystery by examining findings from recent studies of multiple tetraspanins, particularly CD151. The molecular basis for differential impact of tetraspanins on tumor development, progression, and spreading to secondary sites is highlighted, and the complexity and plasticity of their control over tumor cell activities and interaction with their surroundings is discussed. Finally, an outlook is provided regarding tetraspanins as candidate biomarkers and targets for the diagnosis and treatment of human cancer.

**Abstract:**

As a family of integral membrane proteins, tetraspanins have been functionally linked to a wide spectrum of human cancers, ranging from breast, colon, lung, ovarian, prostate, and skin carcinomas to glioblastoma. CD151 is one such prominent member of the tetraspanin family recently suggested to mediate tumor development, growth, and progression in oncogenic context- and cell lineage-dependent manners. In the current review, we summarize recent advances in mechanistic understanding of the function and signaling of integrin-associated CD151 and other tetraspanins in multiple cancer types. We also highlight emerging genetic and epigenetic evidence on the intrinsic links between tetraspanins, the epithelial-mesenchymal transition (EMT), cancer stem cells (CSCs), and the Wnt/β-catenin pathway, as well as the dynamics of exosome and cellular metabolism. Finally, we discuss the implications of the highly plastic nature and epigenetic susceptibility of CD151 expression, function, and signaling for clinical diagnosis and therapeutic intervention for human cancer.

## 1. Introduction

Tetraspanins are a family of integral membrane proteins widely expressed in human tissues and are linked to normal developmental and physiological processes, immunity, and pathologies of many human diseases, including cancer [1,2,3,4,5,6,7]. Structurally, these 20 50 kDa molecules are featured by the presence of highly conserved amino acid residues in each of their extracellular small and large loops, which are connected to four lipid-interacting transmembrane domains and two short cytoplasmic tails [8,9,10]. Despite their high structural similarity, tetraspanins seem to differ both in their functions and signal transduction. Notably, a single point mutation or deletion of the CD151 or CD37 gene leads to kidney and skin malfunctions or defective immunity in both humans and mice, while the targeted deletion or alternative splicing of tetraspanin CD9, CD37, CD53, CD81, and CD82 (singularly or in combination) profoundly impairs function or maturation of B and T lymphocytes and myeloid lineage cells or tissue metabolism in vertebrates [4,7,11,12,13,14,15,16,17,18,19]. Aberrant expression of tetraspanins have been observed in tumor tissues across a wide spectrum of cancer types, ranging from breast, colon, ovarian and prostate cancers to glioblastoma and leukemia [1,20,21]. A series of in vitro and in vivo and clinical studies have highlighted strong roles of these integral membrane proteins in tumor growth, metabolism, angiogenesis, and metastatic dissemination [22,23,24,25,26,27]. Additionally, within the tetraspanin family, CD151 has long been regarded as a potent driver for cell adhesion and behaviors, and tumor onset, growth, angiogenesis and metastatic dissemination, and cancer relapse or resistance to current chemo- and targeted therapies [1,28,29,30,31,32]. 

Intriguingly, there is growing evidence that many tetraspanins such as CD151 and CD82 can negatively or positively influence tumor development and metastasis in oncogenic context- and cell lineage-dependent manners, where they regulate strength of cell-cell junctions, signal transduction of the Wnt pathway, the epithelial-mesenchymal transition (EMT), maintenance of cancer stem cells (CSCs), and dynamics of exosomes (Table 1) [20,21,28,33,34,35]. This emerging paradigm presents a challenge to conceptualize functional and signaling roles of the tetraspanin family in human cancer, and their utilities as biomarkers and drug targets for cancer diagnosis and treatment. In the current review, we will summarize this dilemma for tetraspanin molecules, particularly CD151, and discuss their implications for conceptualizing the role of tetraspanins in human cancer at the cellular, signaling, and epigenetic levels. In addition, our review will be concentrated on the evolving role of CD151 and associated protein complexes in breast, ovarian and prostate cancers, and glioblastoma, as well as their potential for clinical application.

## 2. The Complex Role of CD151 in Tumor Metastasis

### 2.1. Early Studies and Present Views on the Pro-Metastatic Role of CD151

The link between CD151 and human cancer was initially implicated following discovery of its co-translation with laminin-binding α3β1 integrin in human carcinoma cells or modulation of αIIbβ3 function in platelets [6,22,68,69]. The hard evidence of a role of CD151 in cancer, however, came from detection of elevated CD151 expression in metastatic tumor cells and the inhibition of cell motility by an anti-CD151 monoclonal antibody [5]. This pro-metastatic role was subsequently confirmed by a large body of cell line-based in vitro studies in multiple cancer types and has been substantiated by its pro-invasive and pro-angiogenic functions [1,70,71]. More recently, we and others have detected a marked decrease in the formation of pulmonary metastases in breast cancer upon CD151 deletion or knockdown in MMTV-ErbB2 or MMTV-PyMT transgenic or xenograft model [28,32,38]. In line with this evidence, CD151 expression appears elevated at the mRNA level in metastatic tumors and is associated with poor clinical outcomes in the ErbB2^+^ breast cancer subtype [32]. Collectively, these in vitro and in vivo investigations, along with clinical analyses, underpin CD151 as a premier player in human cancer metastasis.

Mechanistically, downregulation or removal of CD151 markedly impairs integrin-dependent tumor cell adhesion, motility, and invasion [21,72]. Studies from our groups and others have also implicated that the pro-metastatic function of CD151 is in part tied to its impact on tumor cell survival or their resistance to anoikis [21,32,71]. Presence of CD151 molecules in tumor cells appears to facilitate the clustering/activation of laminin-binding (LB) integrins (α3β1, α6β1 and α6β4) on the cell surface, which in turn activates SFKs/FAK-, JAK/STAT3-, RAS/MEK- and NF-kB-dependent signaling pathways, cytoskeleton remodeling, and diverse cellular activities and behaviors [32,38,44,71,73]. Interestingly, in breast cancer cells exhibiting a strong promoting role of CD151, receptor tyrosine kinases (RTKs) (e.g., EGFR, ErbB2, c-Met and Ron) or K-Ras are frequently activated because of gene amplification/overexpression or mutations (Figure 1) [1]. 

Part of the pro-metastatic role of CD151 may be linked to its regulation of tumor angiogenesis and microenvironments, as it supports function and integrity of vascular endothelial cells and infiltration of tumor-associated macrophages [25,28,48,74,75]. Consistent with these observations, CD151 deletion markedly decreases expression of several key myeloid cell-associated hallmark genes in mammary tumors, including CD36 and MMP2 [28]. Such notion is also supported by recent findings on the role of CD151-associated α6β4 integrin in macrophages [76]. Additionally, CD151 is implicated to propel tumor progression by regulating integrity and trafficking of exosomes produced in tumor-associated fibroblasts and the Wnt pathway [30,77,78]. In some cancer types, the pro-metastatic role of CD151 appears to be achieved through strong synergy with other tetraspanins (e.g., TSPAN8) [25]. Hence, the impact of CD151 on cancer metastasis largely stems from regulation of tumor cell behaviors and survival, and their microenvironments.

### 2.2. Emerging Evidence of an Anti-Metastatic Role

Unexpectedly, several recent studies have implicated CD151 as a suppressor of tumor metastasis, particularly in ovarian and prostate cancers [46,47,49,79]. A parallel scenario has also been raised for other tetraspanins, such as CD9 and CD82, traditionally regarded as bona fide metastasis suppressors (Table 1). This is a sharp deviation from the well-established paradigm of CD151 being pro-metastatic and CD9/CD82 being anti-metastatic [1]. In fact, the current view of the pro-metastatic role of CD151 in prostate cancer was largely drawn from studies with the endocrine subtype-related Tramp model, and analyses with the PC-3 cell line which is regarded as oncogene-targeting squamous epithelial cells [35,36,46,80]. In contrast, the studies suggesting an anti-metastatic role of CD151 were performed with the oncogene-targeting cells falling into the differentiated epithelial cell category, where CD151 is abundantly expressed at the cell-cell junction [46,47]. In such context, loss of CD151 at cell-cell junctions leads to induction of EMT or a highly motile/invasive mesenchymal cellular phenotype featuring altered expression of typical epithelial (e.g., E-cadherin) and mesenchymal makers (e.g., vimentin, fibronectin and transcription factors Slug/Snail, etc.), enhanced tumor cell motility and invasiveness, as well as strong extracellular matrix (ECM)- arginine-glycine-aspartate (RGD) motif-binding integrin interactions [31,81]. Consistent with this line of observations, data from histological and genomic analyses show that a fraction of human breast and prostate carcinomas may arise from PTEN or E-cadherin mutations or loss in differentiated epithelial cells [46]. It may also be linked to activation of the non-canonical Wnt pathway [28,46,49]. Furthermore, this paradigm may be originated from the cell lineage- and the oncogenic context-dependent role of CD151, highly reminiscent of its associated laminin-binding integrin α3β1 or α6β4 integrin, which seem to vary with cancer subtype or oncogenic context (inactivation/loss of tumor suppressors p53 and SMAD4 versus activation of Ras oncogene) [82,83]. More surprisingly, an opposing scenario appears to occur in a group of tetraspanins traditionally regarded as metastasis suppressors, including CD9 and CD82, in which they seem to dampen or sequester activation or signal transduction of another class of growth factors or receptors, such as the membrane-bound TGF-α or TGF-β Type II receptor [84,85]. 

## 3. A Oncogenic Context- and Cell Lineage-Dependent Role of CD151 in Tumor Growth and Metabolism

Aside from being a key player in tumor metastasis, CD151 has long been regarded as being pro-tumorigenic in multiple cancer types, particularly in breast and skin cancers [28,29,32,86]. Again, this unidirectional view has recently been challenged by data from a series of in vivo studies with CD151 gene-targeted mice, transgenic animal models, and clinical analyses [32,39,71,86]. Because of the strong clinical implications, here we will discuss this twist by centering around recent studies of CD151 function in solid tumors, particularly breast and prostate carcinomas.

### 3.1. Being Pro-Tumorigenic

As one of the most common cancer types among women worldwide, breast cancer is a highly heterogeneous disease. Based on histological and genetic alterations, breast cancer is grossly categorized into four major subtypes: estrogen receptor (ER)-negative, including ErbB2^+^ and triple-negative, and ER-positive (Luminal A and Luminal B) [87]. The malignancy of these breast tumors is largely driven by activation of the PI3K/Akt and RAS/MEK/ERK pathways and loss of key tumor suppressors (p53, BRCA1/2, PTEN, RB, etc.) [87,88,89,90]. In the case of CD151, a series of in vitro and in vivo studies reveal a strong inhibitory effect of CD151 downregulation or deletion on tumor onset and growth in either ErbB2^+^ or basal-like subtypes [29,32,38,71,86]. Collectively speaking, there is robust evidence that CD151 is pro-tumorigenic in the context of signaling driven by overexpression/amplification of RTKs or by the oncogenic activation of their downstream effector pathways (e.g., RAS/ERK1/2 and PI3K/AKT, as well as TGF-β- or Wnt/β-catenin-mediated pathways) (Figure 1 and Figure 2A).

Another developing theme on CD151 in breast cancer is its promoting role in development and growth of basal-like mammary tumors driven by the oncogenic Wnt1 pathway [28]. As a key downstream effector of the canonical Wnt pathway, transcription factor Myc has long been implicated in regulation of the pro-tumorigenic role of CD151 in gliomas [43]. Consistent with this notion, we and others have observed a strong tumor-promoting role of CD151 in glioblastoma [40,42]. Intriguingly, Myc, a key effector downstream of an array of oncogenic pathways, including RAS/Erk and PI3/Akt /mTOR, also serves as a master driver of cell metabolism in diverse cancer types, particularly for nutrients glutamine and glucose [43,90,91,92]. Following this link, we speculate that CD151 might promote tumor growth in breast cancer largely through regulation of c-Myc-driven cellular metabolism. So far, this notion is supported by recent studies of the role of tetraspanins CD9 and CD81 in metabolism of glutamate and lipids in normal or tumor tissues [8,11,27].

Importantly, CD151 appears to be a key player in maintenance of tumor-initiating cells in breast, prostate, and pancreatic cancers, that is, CSCs, as CD151-null/deficient ER^+^ tumor cells seem unable to sustain the population under in vitro culture [28,37,65,93]. This line of observation is consistent with the critical role of CD151-associated α6 integrin in human cancer stem cells across a wide range of human cancer types, particularly breast cancer and glioblastoma [94,95]. They are also of clinical importance, as CSCs are regarded as front runners for candidate therapeutic targets given their crucial roles in drug resistance and disease recurrence in human cancers [96,97].

### 3.2. Being Tumor-Suppressive

In contrast to the role in ER^−^ breast cancer (basal-like and ErbB2+) summarized above, CD151 appears to be tumor suppressive in some cases, for example, during the development and growth of ER^+^ mammary tumors in the mouse mammary tumor virus (MMTV) promoter-driven Wnt1 oncogene model (Figure 2B) [28]. In fact, this unexpected observation is consistent with our prior analysis of the impact of CD151 deletion on mammary luminal progenitor cells (CD24^high^ CD49f^low^ population), where CD151 seems involved in maintenance of quiescence of mammary progenitor cells and the associated impact on mammary gland development [93]. Additionally, it is supported by analyses of CD151 and α3β1 integrin expression in tumor biopsies of lobular or inflammatory breast cancer or colon and ovarian cancer patients [33,49,50,93]. More surprisingly, removing one or two CD151 alleles seems to have nearly an equivalent effect on tumor growth [28], thus illuminating the haploinsufficient nature of CD151 gene and a strong player in human cancer.

Furthermore, the unique aspect of the tumor-suppressive role of CD151 is its intrinsic association with the dynamics of the EMT phenotype, a hallmark trait for the progression of epithelial-origin tumors [32,49]. It has long been advocated that CD151 is a crucial contributor of cell-cell adhesion in immortalized epithelial or carcinoma cells [98,99]. In line with this notion, downregulation or loss of CD151 expression not only weakens such junctional structures, but leads to activation of the transcription factors Snail and Slug and associated EMT-like phenotype (Figure 2B) [28,49]. In line with this functional impact, the downregulation or deletion of CD151 in normal luminal progenitor cells or related breast cancer cell lines or mammary gland is accompanied by the upregulation of fibronectin expression and Slug [93]. While this function has been linked to activation of protein kinase C (PKC) and Cdc42, it remains controversial in terms of integrin involvement [46,47]. 

One of the noticeable observations from our study with the MMTV-Wnt model was the absence of tumor-initiating cells in CD151-deficient ER^+^ mammary tumors [28]. This phenomenon is highly unexpected, since the canonical Wnt pathway or transcription factors Snail and Slug or associated EMT have long been regarded as major drivers of the metastatic progression of ER^+^ breast tumors [94,100,101]. This may also represent another layer of complexity of CD151 action in human ER^+^ breast cancer.

## 4. Molecular Basis for Functional and Signaling Versatility of CD151 and Other Tetraspanins

Mechanistically, the complex role of CD151 in human cancer is intimately linked to diversity of its laterally associated molecular partners on the cell surface, besides heterogeneity in its subcellular localization in tumor cells. Despite lack of extracellular ligands or classical domains/motifs for intracellular protein-protein interactions, tetraspanins are capable of carrying out a variety of functional and signaling roles through at least three distinct types of molecular interactions: (1) Imposing a lateral impact on the activation of their cell surface partners; (2) Recruiting signaling molecules via self-association-based micro- or nano-domains; (3) Long-range impact via regulation of secretory vesicles [73,102,103,104,105]. To date, there is a consensus that tetraspanin molecules, together with their molecular partners, form a nano-scale protein complex or molecular network on the plasma membrane, termed as tetraspanin-enriched microdomain (TEM) [3,8,10,102,106]. 

In CD151-based TEM, there exists at least two distinct pools of tetraspanin molecules on the cell surface. One pool contains large-sized transmembrane protein or receptors through protein-protein interactions, such as LB-integrins interacting with CD151 through their extracellular domains [10]. Another pool, based on our prior biochemical and antibody-based analyses, consists of self-associated or interspecies aggregates of CD151 and other tetraspanins, where they seem localized at the periphery of TEM [103,105,107]. The targeting and stability of this pool of molecules is highly dependent on the palmitoylation of its membrane-proximal cysteine residues and N-terminal cytoplasmic tail [103,104]. Interestingly, the integrin-absent pool of molecular aggregates of CD151, which presumably corresponds to the so-called integrin-free CD151 fraction, seems to have a regulatory role in cell-cell contact and tumor resistance to therapeutic agents [47,108]. More recently, we and others have noted that some of these molecular interactions varies with cell lineage, that is the cell-of-origin or differentiation state of oncogene-targeting cells, as well as oncogenic context (RAS vs. mutation or PTEN mutation) [28,47,49]. Here, we will examine such advances in the context of both pro- and anti-tumorigenic roles of tetraspanins in human cancers.

### 4.1. Hijacking Function and Signaling of Single Transmembrane-Containing Receptor or Protein Partner

Although many tetraspanins are regarded as key mediators of tumor development and progression, their actions have been connected to at least two classes of cell surface molecules: CD151-associated heterodimeric adhesion receptors for laminins, α3β1, α6β1, and α6β4 integrins, and the CD9/CD81/CD82-binding Ig-G-containing proteins such as EWI-2 and EWI-F. In the case of CD151, its pro-tumorigenic role is largely carried out through regulation of function and signaling of laminin-binding integrins (Figure 2A) [32,71]. Studies from our group and others indicate that the pro-metastatic role of CD151 in human basal-like and ErbB2 breast cancer subtypes is achieved largely through regulating α6 integrin-dependent cell motility, invasion and survival [45,50]. In particular, CD151 contributes to the lateral clustering in cis of α6 integrins through direct extracellular domain linkages, which in turn enhances clustering/activation of these adhesion receptors and subsequent changes in cell-ECM adhesion, cytoskeleton remodeling, and signal transduction [71,73,109]. 

The tumor-promoting role of CD151 may also be carried out through regulation of α3β1 integrin-dependent cell-ECM adhesion, migration, survival, and signaling [50,110,111,112]. Such impact, however, seems largely restricted to tumor cells with basal cell lineage or a mesenchymal phenotype [28,71,113]. In contrast, in tumor cells with strong epithelial cell characteristics, CD151, like α3β1 integrin, is more engaged in the maintenance of cell-cell contact through basolateral distribution, conferring an anti-tumor role in an integrin-dependent manner or through interacting non-integrin partners or self-association/clustering [47,49,114]. In this case, CD151 seems to repress tumor cell growth by counteracting EMT in multiple cancer types, including breast, ovarian and prostate [28,46,49,115]. In fact, this scenario highly resembles the well-established established tumor-suppressive role of tetraspanins CD82, CD9, and TSPAN8 in epithelial-origin cancers, whereby they regulate E-cadherin/β-catenin complex-dependent cell-cell adhesion [45,50,99,116,117,118]. For CD9 and CD82, this function appears to take place through EWI proteins [10,20,21,58]. Interestingly, these tetraspanins are capable of suppressing metastasis of mesenchymal cell-origin melanoma by sequestering activation of TGF-β type II receptor [85]. Hence, tetraspanins impact tumor growth and progression largely through regulation of function and signaling of their major partners in a parasitic manner, regardless of being pro- or anti-tumorigenic. 

### 4.2. Recruiting Signaling Molecules via the Tetraspanin Self-Associated Membrane Microdomain

Another important mode for tetraspanin-mediated tumorigenesis is to relay PKC-dependent signaling through formation of tetraspanin-enriched microdomain (TEM) [3,8,10]. Based on biochemical and microscopy-based studies, this type of interaction may involve palmitoylation of multiple membrane-proximal cysteine residues in both N- and C-terminals of tetraspanin molecules, which provide key support for clustering or oligomerization of these molecules on the cell surface (Figure 2) [102,119,120]. Evidence from extensive biochemical and microscopy studies have firmly established that the primary function of TEM is to recruit PKC to TEM-associated protein complexes, molecular aggregates or cluster-like structures on the cell surface [98,120]. In case of CD151, TEM recruits PKC-α to phosphorylate laminin-binding (LB) α3, α6 or β4 integrins, while CD53-mediated TEM drives recruitment of PKC-β to BCR complexes in B cells, which in turn leads to signal transduction [71,73,109]. In this context, the tumor-suppressive role of integrin-free CD151 in prostate cancer [47] may be regarded as recruiting the PKC-like signaling molecules to strengthen cell-cell adhesion through oligomerized CD151 molecules. Consistent with this notion, compared to immortalized mammary basal epithelial cells (MCF-10A), luminal epithelia cells (MCF-7) exhibit relatively poor expression of laminins and LB integrins, while having strong E-cadherin expression and cell-cell interactions [71,103]. A similar scenario can be said for the so-called integrin-free CD151 in prostate cancer cells [47]. Moreover, the TEM assembly may involve multiple intracellular membrane compartments, such as endoplasmic reticulum, Golgi, lysosomal, endosomal and multivesicular bodies, which have been extensively described by multiple reviews [10,20,21].

Another remarkable advance in our understanding of TEM is that it can be visualized as nanometer-sized molecular aggregates on the cell surface using high-resolution microscopy [102,119]. These imaging-based findings are consistent with the observation of molecular composition of TEM from our prior biochemical analyses [71,103,104,114]. This type of approach is of particular value to delineate key molecular components or interactions in TEM across various cancer types, ultimately accelerating our mechanistic understanding of the crucial role of CD151 and other tetraspanins in tumorigenesis and metastasis.

### 4.3. The Long-Range Effect via Regulation of Exosome Formation, Trafficking and Function

Besides TEM, tetraspanins are capable of impacting tumorigenic and metastatic processes through regulation of secretory vesicles named as exosomes [30,121,122]. As a class of extracellular vesicles, exosomes are <200 nm in diameter and formed by cells through invagination of endosomal and plasma membranes. Exosomes are highly enriched in tetraspanins, particularly CD63 and CD151, in addition to diverse intracellular components, including diverse RNA species, metabolites, and proteases [10,121,123]. Given the nature and signaling capabilities of TEMs, it is of no surprise that many tetraspanins are regarded as key contributors to a variety of exosome-associated functional roles, ranging from tumor cell migration, angiogenesis and signal transduction to expression of critical cancer genes and tumor metabolism through control of micro-RNA or non-coding RNA or metabolite pools [67,124,125]. In case of CD151, it is suggested to act in concert with TSPAN8 to drive exosome production in both tumor and endothelial cells, thereby facilitating tumor metastasis [30,118]. However, the mechanism for tetraspanin-dependent regulation of exosomes could be far more complex than originally thought, as some tetraspanins seem to negatively regulate activity and integrity of key exosome-producing machinery, that is, the endosomal sorting complexes required for transport (ESCRT) [126]. With variation in extent of palmitoylation and glycosylation between tetraspanins, they may mediate exosome functions by modulation of their lipid composition [41]. Finally, Wnt signaling has long been known for having a long-range effect on tumor immune microenvironments [77]. Given the strong link between CD151 and exosomes, we speculate that the significant role of CD151 in Wnt-induced mammary tumorigenesis may be partially achieved through regulation of exosome-mediated delivery of Wnt ligands [28]. 

Overall, compared to traditional cell surface molecules or receptors, the impact of CD151 and other tetraspanins on tumor growth and metastasis is more closely linked to regulation of multi-component protein complexes on the cell surface, extracellular vesicles and tumor microenvironments.

### 4.4. Decoding the Myth of the Crosstalk Between Tetrspanins and the Wnt Pathway in Cancer Cells

Our recent in vivo study suggests additional complexity of tetraspanin function in cancer cells, particularly in the context of oncogenic activation of the Wnt-dependent pathway (Figure 2). As one of the widely activated oncogenic pathways, Wnt signaling involves interactions between extracellular ligands (Wnts) and multi-component protein complexes on the cell surface composed of seven transmembrane-spanning Frizzled 1-7 and Type I transmembrane co-receptors (e.g., LRP5/6) [127]. Importantly, many of these components, along with their downstream effectors (i.e., Axin, DVL, β-catenin, and TCF7-L2) or mediators (e.g., RNF43) are frequently overexpressed or downregulated in human cancers, particularly those of epithelial cell origin [128,129,130]. The Wnt pathway can also be constitutively activated through genetic mutations or deletions of their protein destruction complexes such as by APC mutation, accompanied by translocation and elevated transcriptional activity of β-catenin [131]. Additionally, the Wnt pathway has been strongly implicated in tumor recurrence and progression in multiple cancer types [127,132,133,134]. 

Our recent study shows that upon CD151 deletion, there was more than 10-fold increase in the level of nuclear β-catenin as well as the strong cytosolic presence of E-cadherin in mammary tumors [117]. These molecular and signaling changes ultimately bolster transcriptional activation of pro-proliferative genes and tumor growth [135,136]. Also, the genes that regulate cell proliferation, survival, and metabolism (e.g., *Cyclin D1* and *Myb*) are strongly affected [127]. Additionally, expression of the genes involved in regulation of the stability of E-cadherin/β-catenin complexes appeared suppressed, strengthening the intrinsic role of CD151 in cell-cell adhesion [34,116,117,137]. In contrast, CD151 disruption markedly blunts survival of epithelial basal cell-derived tumor cells [28], consistent with the impact of CD151 knockdown on MDA-MB-231 cells [38,71]. Importantly, these emerging observations on CD151 are in line with the current paradigm over the crosstalk between other tetraspanins, including TSPAN8, TSPAN5, and CD82, and the Wnt/β-catenin pathway [1,71,109,138]. Meanwhile, expression of α3 integrin in luminal cells appear unaffected, implicating that the tumor-suppressive role of CD151 in breast, prostate and ovarian cancers are attributed to the combined action of integrin- and self-associated CD151 molecules. As a result, our studies argue that CD151 is a suppressor of ER^+^ breast cancer and prostate cancer, as they frequently arise from oncogenic targeting of luminal or well-differentiated epithelial cells, rather than basal epithelial or progenitor cells. 

## 5. Control of Expression of CD151 and Other Tetraspanins at Multiple Levels

There is evidence that in contrast to traditional oncogenes or tumor suppressors, the role of tetraspanins including CD151 during carcinogenesis and metastasis is achieved largely through altered expression level and associated impact on activation state and signaling strength of their associated receptors or protein complexes in tumor cells. Notably, very few tetraspanins exhibit functional loss/gain due to genetic alterations (mutations, amplifications, or deletions) [1,20,54,60]. In this sense, the impact of tetraspanins on cancer development and progression may primarily stem from their deregulated expression, subcellular distribution, and molecular partners. However, there is accumulating evidence that expression of tetraspanins seems more regulated at transcriptional and epigenetic levels, largely reminiscent of deregulation of classical tumor suppressors, such as PTEN or BRCA1/2 genes [89].

Thus far, DNA hypermethylation in cancer cells has been documented for at least 6 members of the tetraspanin family, including CD9, CD81, CD82, CD151, TSPAN1, TSPAN3, and Uroplakin [46,63,139,140,141,142,143]. Largely occurring in their promoter regions [139,142], DNA hypermethylation ultimately leads to decreased mRNA and/or protein levels of tetraspanins in tumor cells or tissues [10,144,145,146,147]. Interestingly, this type of regulation appears common in prostate, colon, and ovarian carcinomas, as well as in neuroblastomas and glioblastomas [148,149]. The precise mediators behind DNA methylation of tetraspanin genes however, remain to be identified. Based on recent studies, this type of regulation may be particularly evident in cancer driven by MYC-N amplification and activation of inflammation-oriented NF-κB-driven pro-survival network [92], as well as activation of RTK- or TGF-β receptor-mediated oncogenic pathways [71,85,86,150,151].

The deregulated expression of tetraspanins during tumorigenesis and metastasis may also be attributed to dynamics of their associated protein partners at the co-translational level [82,83,152]. In the case of CD151, its downregulation in tumor cells may be associated with decreased protein expression of α3β1 and α6β4 integrins [32]. There may be a similar scenario for the decreased expression of CD9/CD81-associated EWI proteins in aggressive melanoma [85,140,142,153]. Additionally, there is evidence that tetraspanin expression is regulated through proteasome- or microRNA-mediated biochemical processes [1,151]. Combined, the functional plasticity of tetraspanins during tumorigenesis and metastasis is, at least in part, achieved by tight regulation of their expression at epigenetic, transcriptional, and translational levels, and through protein degradation machinery [154].

## 6. Clinical Significance of Deregulation of CD151 and Its Associated Network

Even though CD151 has a complex role in human carcinomas, the tight link between CD151 expression and tumor relapse provide a unique window for pursuing CD151 as a drug target [1,31]. This link is also strongly supported by its role in the differentiation of mammary progenitor cells (maintenance of quiescence) in CD151-targeted mice and the MMTV-Wnt model [155,156]. Additionally, CD151 and CD9 are tightly associated with activities of tumor-initiating cells or CSCs [27,28,37]. Moreover, this potential targeting is bolstered by the wide recognition of the role of CD151-associated α6 integrin in survival or activity of CSCs and may be attributed to the intimate crosstalk between CD151-α6 integrin complexes and the RAS/MEK/ERK pathway in basal epithelia cells [28,82]. In this regard, our observed effect of CD151 deletion is consistent with the role of α6 integrin/CD49f-based CSCs in the MMTV-Wnt1 model described by the Perou group [96]. Because CD151 is associated with recurrence of basal-like breast cancer [31], it will be of interest to determine whether the CSC-associated role of CD151 is recapitulated by use of more clinically relevant PDX model of breast cancer under taxane-based regimens known to foster CSCs [157]. This notion is also supported by the evidence that altered CD151 promotes cancer cell resistance to targeted (anti-ErbB receptors) and chemo- therapies in multiple cancer types [32,40,86,108]. Meanwhile, a number of tetraspanins have been shown to be key players in hematopoietic stem/progenitor cells [15,93,158]. Conversely, the strong role of tetraspanins, such as CD151 and CD37, in cell differentiation could serve as a basis for development of lineage-based targeting, in a manner similar to the antibody targeting of CD20 [159]. In case of CD37, because of its restricted expression and strong biological role in mature B lymphocytes, it has been clinically targeted with monoclonal antibody for leukemia treatment. The efficacy of this approach, however, appears challenged by emerging incidence of mutations at the critical domain or deletion of this gene in biopsies in some patient populations [16].

Meanwhile, the prevalence of DNA methylation in the promoter regions of tetraspanins provides a unique opportunity for the evaluation of tumor progression, as their regulation frequently correlates with the onset of metastases in multiple cancer types. Such data may also provide complimentary support for carcinogenesis in the context of tetraspanin-enriched exosomes in patient body fluids [54]. Tetraspanins are highly regarded as potential therapeutic targets, as they support activities and signaling of cancer cells during angiogenesis and dissimilation to secondary sites of primary tumors [1]. In fact, a number of function-blocking monoclonal antibodies against tetraspanins are under investigation for their anti-tumor efficacy [5]. Also, some tetraspanins have been chosen for launching the chimeric antigen receptor T cell (CAR-T)-based anti-cancer therapy [160]. Meanwhile, the impact of tetraspanins on EMT, CSCs and other malignant processes is susceptible to the epigenetic regulation [34,45,49,98,99,117,118,137,161,162,163]. Thus, targeting DNA hypermethylation through chemical inhibitors of DNA methyltransferases, such as 5-Aza-CdR or HDAC inhibition via Trichostatin A, may provide a means to restore expression of many tetraspanins and their anti-tumor functions in various carcinomas, such as CD82 and CD151 in prostate cancer and CD81 in neuroblastomas or glioblastomas [148,149,163,164,165].

## 7. Conclusions

The role of CD151 and other tetraspanins during tumor growth and progression has long been postulated to be manifested through regulation of tumor cell adhesion, survival, migration, and invasion in an integrin-dependent manner. Especially for CD151, it is regarded as a key adaptor for activation and signal transduction of multiple LB-binding integrins on the cell surface [32,39,45,50,71]. Over the past decade, CD151 is increasingly appreciated as a functionally versatile molecule and driver of human epithelial-origin cancers whereby it regulates cell-cell junctions, proliferation, EMT, metabolism and CSCs, as well as tumor microenvironments [54]. These diverse functions highlight a key basis for the crucial role of CD151 across a wide spectrum of human cancer [1,3]. Additionally, the presence of CD151 in exosomes derived from tumors or their microenvironments suggests a new mechanism for its role in tumor relapse and drug resistance [41,48,54]. There is also evidence on the cell lineage- and oncogenic context-dependent roles of CD151 and other tetraspanins [28,29,48,50,71,93,166]. However, the underlying genetic, epigenetic and metabolic mechanisms remain to be defined. With emerging powerful genomic, proteomic, and metabolomic tools, we will be able to delineate the versatile role of CD151 and other tetraspanins across different stages of cancer development and progression, progression, as well as launch a new line of biomarkers and drug targets for the clinical management of this aggressive disease.

## Figures and Tables

**Figure 1 cancers-13-02005-f001:**
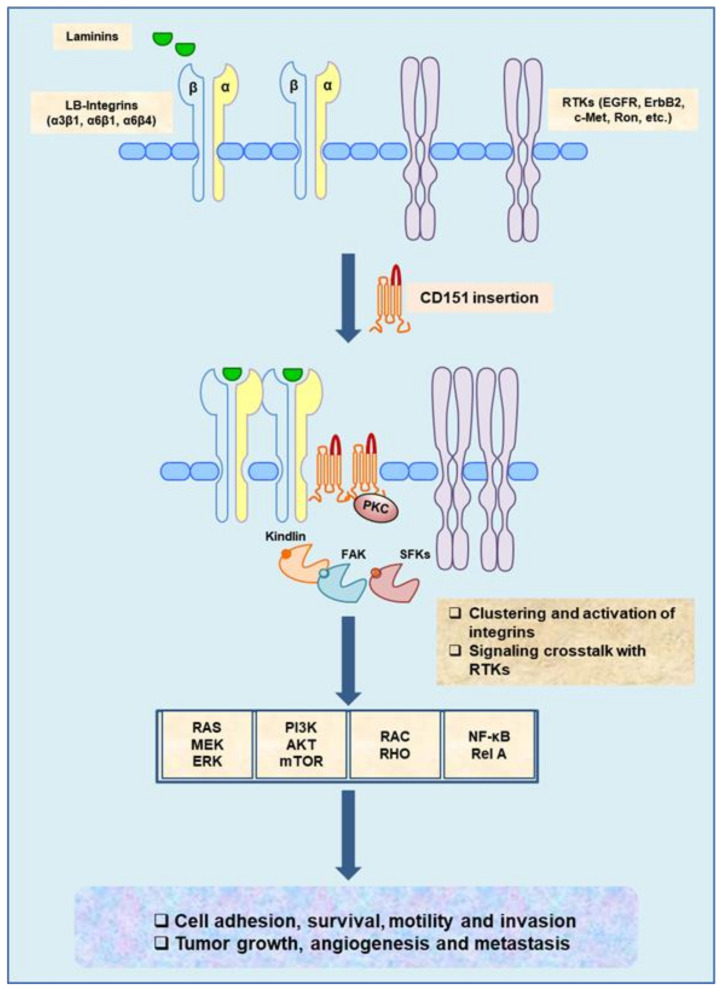
The pro-metastatic function and signaling pathways driven by CD151/laminin-binding (LB) integrin complexes and their crosstalk with oncogenic receptor tyrosine kinases (RTKs) in cancer cells and endothelial cells. The role of CD151 illustrated in a gain-of-function manner is based on observations from a series of studies with gene-targeted or shRNA knockdown effect in mouse models or cancer cell lines.

**Figure 2 cancers-13-02005-f002:**
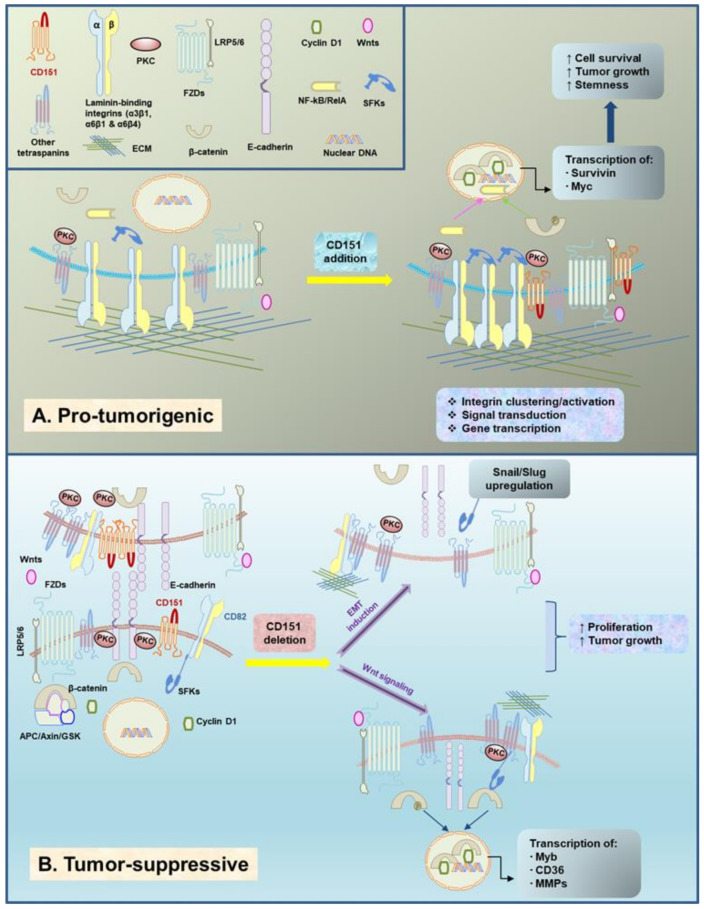
Emerging evidence on the context-dependent crosstalk between tetraspanin CD151 and the Wnt signaling pathway in breast cancer. (**A**) Hypothetical molecular basis for the cooperative role of CD151/laminin-binding integrin complexes and the Wnt/ pathway in basal cell-origin breast cancer. (**B**) A working model for CD151-mediated suppression of the Wnt/β-catenin signaling, and associated changes in transcription factors Snail and Slug, cell-cell adhesion and EMT in luminal epithelial cell-origin breast cancer.

**Table 1 cancers-13-02005-t001:** A glance of tumor-promoting and suppressing roles of tetraspanin molecules across human epithelia-origin cancers *.

Genes	Pro-Malignant Roles	Tumor-Suppressive Roles
*CD151*	Promote prostate cancer malignancy by impacting cancer stem cells and tumor metastasis [36,37]. Promote breast cancer cell proliferation and invasion through regulation of TGF-β1/SMAD pathway [38]. Support tumor invasive behaviors via control of RhoA signaling [39]. Drive glioblastoma malignancy by supporting tumor cell migration, CSCs and exosome release [40,41,42,43]. Promote tumor progression via sustaining collective cell migration and signaling in skin carcinoma [44,45].	Suppress prostate cancer metastasis via regulation of cell-cell contact and EMT [46,47]. Suppress inflammatory breast cancer through regulation of macrophage recruitment [48]. Suppress tumor cell proliferation and invasion in lobular breast cancer subtype and ovarian cancer [49,50]. Trigger anti-tumor immunity in breast cancer [51].
*CD9*	Promote tumorigenesis in breast cancer [52]. Support pancreatic cancer development via regulation of cancer stem cells and glutamine metabolism [27].	Suppress lung cancer metastasis [53].
*CD81*	Promote breast cancer metastasis [2]. Support metastatic progression in breast cancer and osteosarcoma [54,55].	Suppress bladder cancer cell invasion [56].
*CD82*	Promote EMT in liver cancer and metastatic potential of malignant prostate carcinoma PC-3 cell line [35,57].	Suppress tumor cell migration and invasion [58]. Suppress melanoma motility by interfering CD44 splicing [59]. Suppress EMT in prostate cancer [34]. Repress tumor cell motility through control of extracellular vesicle production and architecture of plasma membrane [60].
*TSPAN12*	Promote breast cancer metastasis through the Wnt/β-catenin pathway [61]. Drive impact of tumor-associated fibroblasts on tumor cell invasion [62].	Suppress tumor progression in non-small cell lung cancer (NSCLC) [63].
*TSPAN8*	Support cancer stem cells via Sonic Hedgehog pathway [64,65]. Promote tumor cell invasion and metastasis [25,30,66].	Suppress NSCLC metastasis by affecting activity of extracellular vesicles [67].

*: A small fraction of studies listed are on melanoma or glioblastoma. CSC, cancer stem cell; EMT, epithelial-mesenchymal transition.

## Data Availability

Not applicable.

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
