# Peer review of "The Context-Dependent Impact of Integrin-Associated CD151 and Other Tetraspanins on Cancer Development and Progression: A Class of Versatile Mediators of Cellular Function and Signaling, Tumorigenesis and Metastasis"

_cancers, 2021, doi:10.3390/cancers13092005_

Round 1
Reviewer 1 Report
This review documents the function of tetraspanin CD151 in cancer development and metastasis. Tetraspanins function through lateral (in cis) interactions with a variety of different membrane proteins and thereby control cell adhesion, migration, proliferation and signaling. Aberrant expression of tetraspanins has been observed across a wide spectrum of cancer types, ranging from breast, colon, ovarian and prostate cancers to glioblastoma and leukemia. This review focusses on the paradigm that tetraspanins (CD151) can have pro-tumorigenic and anti-tumorigenic roles, depending on the cancer type, context- and cell lineage-dependent manners.
The topic is of high relevance since the involvement of tetraspanins in cancer has been well-established, but underlying mechanisms are still understudied. In particular the observations that a single tetraspanin (CD151) can have such opposing effects in cancer biology has puzzled researchers for many years. The importance of the topic is also underlined by the (pre-)clinical studies that are currently undertaken to develop new cancer therapies based on targeting tetraspanins. Together, this makes the review relevant and timely (no recent reviews on this paradigm).
Different suggestions to improve the manuscript:
Throughout the review CD151 is being linked to downstream signaling pathways. The review would benefit from better explanation of what happens at the cell membrane. Here a more detailed explanation of the concept of tetraspanin-enriched microdomains (TEMs) is needed. Recent advanced microscopy studies have provided novel insight into their composition, mobility and size. Since they are nanometer-scale sized clusters, the term ‘’tetraspanin nanodomains’’ would be more appropriate than ‘’TEMs’’ (reviewed in Deventer et al Trends Cell biology 2021). More evidence is accumulating that specific tetraspanin-protein interactions in the plasma membrane underlie specificity vs redundancy in tetraspanin function. Thus, what is known about different interacting proteins of CD151 (beyond integrins) in different tumors that may explain pro-tumorigenic vs suppressive functions?
It has been reported that CD151 also exists in integrin-free complexes in tumor membranes that affects tumor cell motility and cancer drug resistance (Palmer et al Cancer Res 2014, Hwang et al Cell Mol Life Sci 2019). This should be discussed. Can the authors discuss whether composition of different CD151-complexes (w/o integrins or other binding partners) determines pro/anti-tumorigenic functions of tetraspanins?
Figure 2 is not self-explanatory for the reader. Why are integrins lacking here? What does CD151 actually do? How does CD151 couple to this downstream pathways? If findings are not well-established please remove from the figure or present in another way. Change colors/layout and include proper legend.
What are open questions in the field?
Minor/textual suggestions:
Page 2 ‘’Paradoxically, there is growing evidence that many tetraspanins impact…’’ The paradox is not well explained here. It is only further in the text when readers understand what is meant with the paradox (the pro- and anti-tumorigenic functions of tetraspanins).
Page 4: It is mentioned that CD151 controls tumor cell proliferation and invasion through regulation of TGF-β1/SMAD pathway. Can the authors include more information/ideas on proposed mechanisms: does CD151 interact with TGF-β1 receptors, or is this possibly mediated via integrins?
Page 7: ‘’Additionally, tetraspanins may alter exosome functions by modulation of their lipid composition [103]’’ Please provide correct references for this statement.
P8 ‘’the impact of tetraspanins on cancer development and progression largely stems from their deregulated expression, rather than genetic alterations.’’ Although correct, there are different papers that document genetic alterations in tetraspanins (CD151, CD37, CD81) leading to disease/cancer that would be relevant to include in the review. In addition, studies characterizing splicing variants of tetraspanins are ongoing (Hochheimer, Lang Sci rep 2019)
Page 8: CD151 expression not only regulated at the (epi)-genetic level, but also at the cell surface by protein stabilization (Lineberry et al JBC 2008), trafficking/internalization (Liu et al JBC 2007).
Page 9: discussion on tetraspanins as novel therapeutic targets. It is relevant to include CD37 here considering this is only tetraspanin that has reached the clinic thus far (see many papers on CD37-targeting antibodies for treatment of lymphoma).
Author Response
RESPONSE TO REVIEWER 1 COMMENTS
Reviewer #1: This review documents the function of tetraspanin CD151 in cancer development and metastasis. Tetraspanins function through lateral (in cis) interactions with a variety of different membrane proteins and thereby control cell adhesion, migration, proliferation and signaling. Aberrant expression of tetraspanins has been observed across a wide spectrum of cancer types, ranging from breast, colon, ovarian and prostate cancers to glioblastoma and leukemia. This review focusses on the paradigm that tetraspanins (CD151) can have pro-tumorigenic and anti-tumorigenic roles, depending on the cancer type, context- and cell lineage-dependent manners.
The topic is of high relevance since the involvement of tetraspanins in cancer has been well-established, but underlying mechanisms are still understudied. In particular the observations that a single tetraspanin (CD151) can have such opposing effects in cancer biology has puzzled researchers for many years. The importance of the topic is also underlined by the (pre-)clinical studies that are currently undertaken to develop new cancer therapies based on targeting tetraspanins. Together, this makes the review relevant and timely (no recent reviews on this paradigm).
Response: We thank the reviewer for the extensive, constructive and knowledgeable input and comments to our current manuscript. The following is our point-to-point response to the reviewer’s critique:
- Different suggestions to improve the manuscript:
Throughout the review CD151 is being linked to downstream signaling pathways. The review would benefit from better explanation of what happens at the cell membrane. Here a more detailed explanation of the concept of tetraspanin-enriched microdomains (TEMs) is needed. Recent advanced microscopy studies have provided novel insight into their composition, mobility and size. Since they are nanometer-scale sized clusters, the term ‘’tetraspanin nanodomains’’ would be more appropriate than ‘’TEMs’’ (reviewed in Deventer et al Trends Cell biology 2021).
Response: To address the reviewer’s input, the biochemical and functional/signaling nature of TEM was discussed in section 4, which focuses on molecular mechanism or basis for function and signaling of CD151 in human cancer. We acknowledge in the text the recent finding that the size of TEM is at the range of nanometer scale. However, we still apply the term TEM, rather than TNM, to describe the tetraspanin-enriched molecular aggregates or clusters on the cell surface in the current review. This is largely because the biochemical and signaling nature of these structures remains unchanged, even though their sizes might be at the nano-scale, rather than micro-scale.
- More evidence is accumulating that specific tetraspanin-protein interactions in the plasma membrane underlie specificity vs redundancy in tetraspanin function. Thus, what is known about different interacting proteins of CD151 (beyond integrins) in different tumors that may explain pro-tumorigenic vs suppressive functions?
Response: We agree with reviewer that the specific tetraspanin-protein interactions in the plasma membrane might be the key reason behind specificity and redundancy of tetraspanin functions in tumor cells. Regarding the contrasting role of CD151 in basal-like and ER+ breast cancer (pro- and anti-tumorigenic), we believe it is largely attributed to the differences in subcellular localization of CD151 and associated complexes between basal and luminal epithelial cells. While this part of question is under heavy investigation in my lab, we still provide some insight in the context of diversity and complexity of cell-cell junction in basal and differentiated epithelial cells. In particular, the primary partners of CD151 in basal cells are laminin-binding integrins, other tetraspanins, as well as some key components of the Wnt receptor complexes. In luminal epithelial cells (also ER+), the key interacting partners of CD151 may include E-cadherin and PKC, and some components of non-canonical Wnt complexes. While this part of work is still ongoing in our lab, we highlight the above possibility in our modified version of Figure 2.
- It has been reported that CD151 also exists in integrin-free complexes in tumor membranes that affects tumor cell motility and cancer drug resistance (Palmer et al Cancer Res 2014, Hwang et al Cell Mol Life Sci 2019). This should be discussed
Response: These two studies were discussed in the context of molecular basis (section 4) of our manuscript.
- Can the authors discuss whether composition of different CD151-complexes (w/o integrins or other binding partners) determines pro/anti-tumorigenic functions of tetraspanins?
Response: To address the reviewer’s question, we added a sub-section on the Wnt/CD151 interaction in section 4, where we put particular emphasis on the differences in CD151-assocaietd protein complexes between basal and luminal epithelial cells, which might dictate its contrasting role in breast cancer subtypes or other cancers, such as prostate cancer.
- Figure 2 is not self-explanatory for the reader. Why are integrins lacking here? What does CD151 actually do? How does CD151 couple to these downstream pathways? If findings are not well-established please remove from the figure or present in another way. Change colors/layout and include proper legend.
What are open questions in the field?
Response: In our original Figure 2, we intended to highlight the tumor-suppressive role of CD151 in luminal epithelial cell-originated breast cancer in the context of Wnt oncogenic activation, and associated induction of EMT-like phenotype. In such context, change in E-cadherin is key to CD151 regulation, rather than LB integrins. Also, LB integrins, in general, are poorly expressed compared to their counterparts in basal-like epithelial cells. Given these circumstances, we added a new additional panel of cartoon depicting the pro-tumorigenic role of CD151. In addition, we added PKC as a possible missing link for CD151-mediated downstream signaling pathways in basal cell and luminal cell-derived breast cancer. Furthermore, to make this figure more self-explanatory, we added legends for key individual components of protein complexes.
As far as concern of the pivotal role of CD151 in human cancer, the open question is how it acts as a tumor suppressor through regulation of cell-cell junctions for the following reasons:
- There is strong evidence that CD151 is a potent tumor suppressor of human cancer arising from differentiated epithelial cells, such as some breast cancer subtype and prostate cancer. However, the key underlying mechanisms remain largely elusive.
- The component of cell-cell junction structures varies between cancer types or at different stages of tumorigenesis and metastasis.
- The cell-cell junction structures are highly enriched in components of non-canonical Wnt signaling complexes, a large number of tetraspanins-like molecules, such as claudins and connexins. The functions and signaling of many of these molecules seem to be overlapping with tetraspanins including CD151. Also, they undergo extensive palmitoylation and are able to form aggregates or associated with other molecules to recruit signaling molecules to convey tumor-suppressive functions across a variety of human carcinomas. Importantly, many components in such subcellular structures play a strong tumor-suppressive role.
Minor/textual suggestions:
Page 2 ‘’Paradoxically, there is growing evidence that many tetraspanins impact…’’ The paradox is not well explained here. It is only further in the text when readers understand what is meant with the paradox (the pro- and anti-tumorigenic functions of tetraspanins).
Response: We originally refer to paradoxically as CD151 being negative and positive regulators of breast cancer. In the revised version, we replaced this word with intriguingly.
Page 4: It is mentioned that CD151 controls tumor cell proliferation and invasion through regulation of TGF-β1/SMAD pathway. Can the authors include more information/ideas on proposed mechanisms: does CD151 interact with TGF-β1 receptors, or is this possibly mediated via integrins?
Response: We believed it involves concomitant regulation of laminin-binding integrins, other tetraspanins and sequestering of type II TGF-β1 receptors, which are inserted in the section 4.
Page 7: ‘’Additionally, tetraspanins may alter exosome functions by modulation of their lipid composition [103]’’ Please provide correct references for this statement.
Response: The original references have been replaced with the correct references.
P8 ‘’the impact of tetraspanins on cancer development and progression largely stems from their deregulated expression, rather than genetic alterations.’’ Although correct, there are different papers that document genetic alterations in tetraspanins (CD151, CD37, CD81) leading to disease/cancer that would be relevant to include in the review. In addition, studies characterizing splicing variants of tetraspanins are ongoing (Hochheimer, Lang Sci rep 2019)
Response: We revised our text to highlight the impact of genetic alterations of these tetraspanins, although their impact frequently occurs in non-cancer diseases.
Page 8: CD151 expression not only regulated at the (epi)-genetic level, but also at the cell surface by protein stabilization (Lineberry et al JBC 2008), trafficking/internalization (Liu et al JBC 2007).
Response: These references are added to the text.
Page 9: discussion on tetraspanins as novel therapeutic targets. It is relevant to include CD37 here considering this is only tetraspanin that has reached the clinic thus far (see many papers on CD37-targeting antibodies for treatment of lymphoma).
Response: The information of CD37-based clinical study or trial is added.
Reviewer 2 Report
This comprehensive and well written review describes current knowledge about the complex roles that tetraspanins, in particular CD151, play in tumor progression and metastasis. I have no major weaknesses with the review text as written. However, Figure 2 is quite confusing and should be clarified with a more detailed figure legend. It is not clear what this figure is meant to convey, reading it from left to right. Is this a depiction of loss of cell-cell adhesion, and its impact on Wnt signaling? Some of the cell surface molecules are not labelled (interns, RTKs?).
Author Response
RESPONSE TO REVIEWER 2 COMMENTS
Reviewer #2: This comprehensive and well written review describes current knowledge about the complex roles that tetraspanins, in particular CD151, play in tumor progression and metastasis. I have no major weaknesses with the review text as written. However, Figure 2 is quite confusing and should be clarified with a more detailed figure legend. It is not clear what this figure is meant to convey, reading it from left to right. Is this a depiction of loss of cell-cell adhesion, and its impact on Wnt signaling? Some of the cell surface molecules are not labelled (interns, RTKs?).
Response: We thank the reviewer for the encouraging comments and constructive input. As far as Figure 2, it was originally used to describe the role of CD151 in maintenance of cell-cell contact and its relevance to the Wnt oncogene-induced tumorigenesis. To address the reviewer’s concerns, we have added a new panel to highlight the pro-tumorigenic role of CD151 in basal cell-derived breast cancer. In addition, we added a panel to indicate the legend for Wnt signaling pathway, as well as other cell surface molecules such as integrins and RTKs.
Reviewer 3 Report
The Flipping role of Integrin-Associated CD151 and other tetraspanins in cancer development and progression: A Class of plastic mediators of cellular function and signaling, tumorigenesis and metastasis. By Erfani et al.,
The precise biochemical roles of tetraspanins is yet to be clearly defined in literature and so the present review was an attempt to clarify their role in tumor progression. The authors emphasized the plasticity of these proteins particularly CD151 in the progression of different tumor types. The review filled some missing gaps in our general understanding but unfortunately the information as presented in this review was rather confusing and vague at times. For example Fig. 2 only gave part of the story they wanted to convey. It should convey the pro-tumorigenic and tumor suppressor roles, with a detailed figure legend to help the reader follow the flow of information. The title of the review also needs to be changed. The word "FLIPPING" should be eliminated from the title. Tetraspanins are known to be scaffolding proteins/glycoproteins that organize other membrane proteins into microdomains the are specialized for adhesion and signaling. It would be better to reorganize the review around this theme so that the reader can follow their logic and address the missing links.
Author Response
RESPONSE TO REVIEWER 3 COMMENTS
Reviewer #3: The precise biochemical roles of tetraspanins is yet to be clearly defined in literature and so the present review was an attempt to clarify their role in tumor progression. The authors emphasized the plasticity of these proteins particularly CD151 in the progression of different tumor types. The review filled some missing gaps in our general understanding but unfortunately the information as presented in this review was rather confusing and vague at times. For example Fig. 2 only gave part of the story they wanted to convey. It should convey the pro-tumorigenic and tumor suppressor roles, with a detailed figure legend to help the reader follow the flow of information.
Response: We thank the reviewer for the encouraging comments and constructive input. As far as Figure 2, it was originally used to describe the role of CD151 in maintenance of cell-cell contact and its relevance to the Wnt oncogene-induced tumorigenesis. In following the reviewer’s suggestion, we added a new panel to highlight the pro-tumorigenic role of CD151 in basal cell-derived breast cancer. In addition, we added a legend to highlight key components in our models, including those for the Wnt signaling pathway, as well as other cell surface molecules such as integrins and RTKs.
- The title of the review also needs to be changed. The word "FLIPPING" should be eliminated from the title.
Response: The title has been changed to: “The Context-Dependent Impact of Integrin-Associated CD151 and Other Tetraspanins on Cancer Development and Progression: A Class of Versatile Mediators of Cellular Function and Signaling, Tumorigenesis and Metastasis”.
- Tetraspanins are known to be scaffolding proteins/glycoproteins that organize other membrane proteins into microdomains the are specialized for adhesion and signaling. It would be better to reorganize the review around this theme so that the reader can follow their logic and address the missing links
Response: We agree with the reviewer that the key molecular role of tetraspanins is to serve as a scaffold for assembly of large protein or signaling complexes on the cell surface to impact cell adhesion and other cellular processes or signal transduction. In case of CD151, it seems more engaged with regulation of integrin-dependent cell adhesion, migration, survival and proliferation, in addition to the role described for many other tetraspanins. To address the reviewer’s concern, we have extensively modified the section for molecular basis of CD151 function and signaling in human cancer. In particular, we discussed in detail about the role for tetraspanins as a scaffold for signal transduction through TEM in cancer cells.
Round 2
Reviewer 1 Report
The authors have done a great job in improving this extensive review on CD151 in cancer. The different pathways regulated by CD151 are much better explained and accompanying figures have improved significantly.
The title is very long, it can be considered to shorten the title into: ''The Context-Dependent Impact of Integrin-Associated CD151 and Other Tetraspanins on Cancer Development and Progression''. But this may be a matter of personal preference.
Reviewer 3 Report
The authors addressed all my concerns in their revised version. I therefore now recommend this review for publication.